# Effects of Grape Pomace Polyphenols and In Vitro Gastrointestinal Digestion on Antimicrobial Activity: Recovery of Bioactive Compounds

**DOI:** 10.3390/antiox11030567

**Published:** 2022-03-16

**Authors:** Giusy Rita Caponio, Mirella Noviello, Francesco Maria Calabrese, Giuseppe Gambacorta, Gianluigi Giannelli, Maria De Angelis

**Affiliations:** 1National Institute of Gastroenterology “Saverio de Bellis”, Research Hospital, Castellana Grotte, 70013 Bari, Italy; giusy.caponio@irccsdebellis.it (G.R.C.); gianluigi.giannelli@irccsdebellis.it (G.G.); 2Department of Soil, Plant and Food Sciences, University of Bari Aldo Moro, Via Amendola 165/A, 70126 Bari, Italy; mirella.noviello@uniba.it (M.N.); francesco.calabrese@uniba.it (F.M.C.); giuseppe.gambacorta@uniba.it (G.G.)

**Keywords:** antioxidants, bioactive compounds, antimicrobial activity, in vitro gastrointestinal digestion, probiotics

## Abstract

Grape pomace (GP), a major byproduct obtained from the winemaking process, is characterized by a high amount of phenolic compounds and secondary plant metabolites, with potential beneficial effects on human health. Therefore, GP is a source of bioactive compounds with antioxidant, antimicrobial, and anti-inflammatory activity. As people are paying more attention to sustainability, in this work, we evaluate two different extractions (aqueous and hydroalcoholic) of GP bioactive compounds. In vitro simulated gastrointestinal digestion of the GP extracts was performed to improve the bioavailability and bioaccessibility of polyphenols. The antioxidant activity (ABTS and DPPH assays) and the phenolic characterization of the extracts by UHPLC-DAD were evaluated. The antimicrobial effects of GP antioxidants in combination with a probiotic (*Lactiplantibacillus plantarum*) on the growth of pathogenic microorganisms (*Escherichia coli*, *Bacillus megaterium,* and *Listeria monocytogenes*) were evaluated. As a result, an increase of antioxidant activity of aqueous GP extracts during the gastrointestinal digestion, and a contextual decrease of hydroalcoholic extracts, were detected. The main compounds assessed by UHPLC-DAD were anthocyanins, phenolic acids, flavonoids, and stilbenes. Despite lower antioxidant activity, due to the presence of antimicrobial active compounds, the aqueous extracts inhibited the growth of pathogens.

## 1. Introduction

The winemaking process generates large amounts of solid waste and byproducts, such as vine shoots, stalks, grape pomace (GP), wine lees, and wastewater [1]. GP, the residue of fermented and crushed grapes, is the most abundant winemaking waste [2]. In fact, 1 kg of GP is produced for each 6 L of wine [3] and it is used in the production of alcoholic beverages through distillation, of dyes, for land spreading, as fertilizer, and for animal feed [4]. Considering the phenolic composition of GP, alternatives do exist for other innovative and emerging uses that involve the cosmetic, biomedical, nutraceutical [5], and food sectors, in the production of functional foods [6].

Recently, the proximate composition of GP, characterized by fibers, colorants, minerals, and polyphenolic compounds, was summarized [6,7,8]. Grape polyphenols belong to different classes of compounds, among which phenolic acids, anthocyanins, flavanols, and stilbenes [9,10], still persist (for, approximately, 70%) in GP, after the winemaking process [11]. Grape varieties, vintage, winemaking techniques, and many other factors impact phenolic GP content, leading to a non-homogeneous distribution of compounds [2,12,13,14]. Depending on the chosen oenological practice, the maceration phase differently influences the phenolic content of both wine and GP [13]. These compounds are known for their beneficial effects (e.g., anti-inflammatory [15,16], antiaging [5,9], anticancer [17], cardioprotective [18], antimicrobial, antioxidant [19], and anti-inflammatory properties [20]), on human health.

However, polyphenol bioavailability depends, in turn, on other properties, including: (i) the relative content of compounds released from the food matrix along the digestive system (bioaccessibility), (ii) the digestive stability, and iii) the efficiency of the transepithelial passage (intestinal absorption) [21]. Therefore, compounds capable of tolerating gastrointestinal tract conditions are potentially bioavailable in regard to exerting beneficial effects on the human body [22].

Microbiota is known to influence the absorption of dietary polyphenols in the small intestine [23,24]. Polyphenols, as derived from microbial metabolism, act as prebiotic-like molecules that can modulate the growth of specific bacterial strains and contribute toward maintaining a healthy and resilient microbiota, counteracting the onset of dysbiosis or lifestyle-related disease status [25,26]. Importantly, once there is downstream biotransformation as a result of microbial metabolism, some polyphenols may exert anti-inflammatory activity that delays the onset and/or progression of different gastrointestinal pathologies [27]. Several studies have reported on the antimicrobial activity of phenolic food extracts against bacterial taxa [28,29]. However, no study has reported on the synchronic effect of phenolic compounds and probiotics on these bacteria.

Through our analyses, we fulfilled our primary objective of performing an extensive characterization of the phenolic composition with an evaluation of the antioxidant activity of GP extracts (both aqueous and hydroalcoholic) obtained from Aglianico and Nero di Troia red winemaking. Secondly, in order to deeply inspect the bioaccessibility and stability of the main polyphenols present in extracts, we delved into the antibacterial effects of GP extracts by following an in vitro gastrointestinal digestion process.

## 2. Materials and Methods

### 2.1. Chemicals and Reagents

Gallic acid, syringic acid, quercetin-3-O-glucoside, caftaric acid, isorhamnetin, and myricetin were purchased from Sigma-Aldrich (St. Louis, MO, USA); trans-resveratrol from United States Pharmacopeia (USP, Rockville, MD, USA); kaempferol, ε-viniferin, malvidin-3-O-glucoside, quercetin, rutin hydrate, and (+)-catechin from phyproof^®^ (PhytoLab, Dutendorfer, Germany).

Ethanol for residual analysis—acetonitrile HPLC grade was purchased from Sigma-Aldrich (St. Louis, MO, USA) and formic acid HPLC grade from Muskegon (MI, USA); ABTS (2,20-azino-bis(3-ethylbenzothiazoline-6-sulphonic acid) diammonium salt) was purchased from Sigma Aldrich (Darmstadt, Germany) and DPPH (2,2-diphenyl-1-picrylhydrazyl) was purchased from Sigma Aldrich (Darmstadt, Germany). α-amylase, pepsin, and pancreatin were purchased from Sigma-Aldrich Chemistry (St. Louis, MO, USA), and bile salts were purchased from Oxoid (Hampshire, UK).

### 2.2. Grape Pomace Sampling

The research was conducted in October 2020 on Aglianico and Nero di Troia grape cultivars from two different vineyards in the Corato area (Puglia Region, Italy). GPs were obtained as byproducts of different red winemaking processes performed at an experimental vinery of the Department of Soil, Plant, and Food Sciences (Di.S.S.P.A), University of Bari Aldo Moro. Grapes of Aglianico and Nero di Troia cultivar underwent four winemaking processes:(1)C, control: 5 days of maceration at 25 °C, 2 punching down per day, with the addition of potassium metabisulphite (6 g/hL), yeast (*Saccharomyces cerevisiae* var. *Bayanus*, LALVIN R2™, 20 g/hL), and yeast activator;(2)T, toasted: as control, with the addition of toasted vine-shoot chips in maceration (12 g/L);(3)BT, boiled-toasted: as control, with the addition of boiled-toasted vine-shoot chips in maceration (12 g/L);(4)O, oak: as control, with the addition of oak chips in maceration (12 g/L).

The addition of vine-shoot and oak chips in maceration was carried out in order to verify their impacts on the phenolic compound enrichments in the wine and/or grape pomaces. For this purpose, vine-shoots employed to produce the chips were taken from Primitivo cultivar vines and conditioned for 6 months at room temperature in darkness. Subsequently, vine-shoots were ground into small chips (2–20 mm) using a hammer mill (Dietz-Motoren KG, Dettingen unter Teck, Germany). One-half was toasted at 180 °C for 45 min using a thermostatic oven (TFC 120 forced air oven, ArgoLab, Carpi (MO), Italy), whereas the other half was boiled in water for 10 min and then toasted under the same conditions. Oak chips were from strong toasted French Quercus (I-OAK, Dello, BS, Italy). At the end of maceration, the GP was separated from the wine using a stainless-steel hydro-press, inserted in a plastic bag, and immediately placed at −20 °C until analysis. The moisture content of the GP was measured using a thermobalance (Ladwag MAC 110/NP, Radwag, Poland).

### 2.3. Phenol Extract Preparation

Phenol compounds were extracted from GPs using two different solvents: water (W, aqueous extraction) and ethanol-water (S, hydroalcoholic extraction). Aqueous extracts were prepared according to Kamiloglu and Capanoglu (2014) [30] with slight modifications. Briefly, GP was mixed with distilled water (1:2, *w*/*v*) and submitted to intense agitation with a stomacher 400 lab blender (Seward Medical, London, UK) for 180 s. The extracts were recovered by filtration using filter paper (Cordenons, PN, Italy), followed by a nylon filter (pore size 0.45 mm, Sigma, Ireland), and then stored at −20 °C until analysis.

Hydroalcoholic extracts were obtained from GP samples by solid–liquid extraction following the procedure reported in Caponio et al. (2020) [31], with some modifications. Briefly, 3 g of GP was mixed with ethanol 80% (1:10 *w*/*v*), vortexed for 10 min, sonicated for 15 min (Elmasonic S 60 H, ELMA, Singen, Germany), and finally centrifugated at 12,000× *g* for 10 min (SL 16R Centrifuge, Thermo Scientific, Waltham, MA, USA) to recover the hydroalcoholic extract. Extractions were repeated twice more with 30 mL of 80% ethanol. The three extracts were combined, filtered as above reported, and stored at −20 °C until analysis. All extracts were prepared in triplicate.

### 2.4. In Vitro Gastrointestinal Digestion of the GP Extracts

The GP aqueous extracts immediately followed the in vitro gastrointestinal digestion, whereas the hydroalcoholic extracts were previous evaporated and lyophilized. Briefly, the ethanolic phase of hydroalcoholic extracts was evaporated using a rotary evaporator (IKA^®^ HB 10, VWR International, Radnor, PA, USA) at 40 °C and the remaining aqueous phase was lyophilized using a freeze-dryer (LIO-5P, Cinquepascal SRL, Trezzano, Italy) at a 0.01 bar pressure and a condenser temperature of −50 °C. Then, pellets were resuspended in water and used for the analysis of in vitro gastrointestinal digestion.

Simulation of the effect of the gastrointestinal digestion tract upon the GP extract was performed following the method described by Kamiloglu and Capanoglu (2014) [30], with slight modifications. The in vitro gastrointestinal digestion was performed, comprising of a pepsin-HCl digestion for 3 h at 37 °C (to simulate gastric digestion) and a pancreatin digestion with pancreatin and bile salts for 3 h at 37 °C (to simulate small intestinal digestion).

Briefly, 10 mL of each extract was added to α-amylase (56 mg/mL) (Sigma-Aldrich Chemistry, St. Louis, MO, USA) and to 10 mL of pepsin solution composed by NaCl 125 mM/L + KCl 7 mM/L + NaHCO_3_ 45 mM/L + pepsin (Sigma-Aldrich Chemistry, St. Louis, MO, USA) 3 g/L. Thus, the pH was adjusted to 2, using HCl, and incubated at 37 °C for 180 min in a water bath under shaking. After incubation, an aliquot of gastric-digested extracts was kept and stored at −20 °C before analysis and the remainder was added in equal volume to an intestinal solution. The intestinal solution was simulated by dissolving 0.1 g/100 mL of pancreatin (Sigma-Aldrich Chemistry, St. Louis, MO, USA) and 0.15 g/100 mL bile salts (Oxoid^™^, Hampshire, UK). The pH was adjusted to 8, using NaOH and incubated at 37 °C for 180 min in a water bath under shaking. After incubation, an aliquot of intestinal-digested extract was kept and stored at −20 °C before analysis. All extracts were then filtered using 0.45 μm Whatman filter paper and then analyzed for antioxidant activity and quantitative UHPLC-DAD analysis of phenolic compounds. In summary, Table 1 lists all of the analyzed sample extracts.

### 2.5. Antioxidant Activity

The antioxidant activities of the GP extracts were measured by ABTS and DPPH assays, as reported by Caponio et al. (2020) [31]. The DPPH (2,2-diphenyl-1-picrylhydrazyl) assay was performed by preparing a solution of DPPH 0.08 mM in ethanol. In cuvettes for spectrophotometry, 50 µL of each sample was added to 950 µL of DPPH solution. After 30 min in the dark, the decrease of absorbance was measured at 517 nm using a Cary 60 spectrophotometer Agilent (Cernusco, Milan, Italy). The ABTS [2,2′-azino-bis(3-ethylbenzothiazoline-6-sulfonic acid)] radical was generated by a chemical reaction with potassium persulfate (K_2_S_2_O_8_). Briefly, 25 mL of ABTS (7 mM in H_2_O) was spiked with 440 μL of K_2_S_2_O_8_ (140 mM) and kept in the dark at room temperature for 12–16 h. The working solution, by diluting with H_2_O, was prepared to obtain a final absorbance at 734 nm equal to 0.80 ± 0.02 [32]. The decrease of absorbance was measured at 734 nm after 8 min of incubation. Results were expressed as μmol Trolox equivalents (TE)/g of dry GP. Each sample was analyzed in triplicate.

### 2.6. UHPLC-DAD Phenolic Analysis of GP Extracts and Recovery Index (RI)

The phenol compositions of GP extracts were determined by UHPLC Ultimate 3000 RS Dionex system (Thermo Fisher Scientific, Waltham, MA, USA) composed by LPG-3400RS quaternary pump, WPS-3000 TRS autosampler, TCC-3000RS column older, and PDA-3000RS detector. The analytical separation of compounds was achieved using a Hypersil Gold aQ C18 column (100 × 2.1 mm, 1.9 μm particle size), held at 30 °C, and at a constant flow of 0.3 mL/min with water–formic acid (90:10, *v*/*v*) (solvent A) and acetonitrile–formic acid (99.9:0.1, *v*/*v*) (solvent B). The gradient program of solvent A was as follows: 0–20 min from 98% to 30%; 20–24 min isocratic at 30%, then equilibration at the initial conditions for 9 min. The diode array detector was set at an acquisition range of 220–600 nm. The injection volume of the extracts (previously filtered at 0.22 µm) was 5 μL. Data were acquired and processed using Xcalibur v. 2 (Thermo Fischer Scientific). The identification of the compounds was carried out by comparing the retention times and the spectral parameters of peaks with those of the standards. Specifically, (+)-catechin, kaempferol, rutin hydrate, quercetin-3-O-glucoside, quercetin, isorhamnetin, and myricetin for semi-quantification of flavonoids; gallic acid, caftaric acid, and syringic acid for semi-quantification of phenolic acids; ε-viniferin and *trans*-resveratrol for semi-quantification of stilbenes; malvidin-3-O-glucoside for semi-quantification of anthocyanins were used. Quantitative analysis was performed according to the external standard method based on calibration curves obtained by injecting different concentrations of standard solutions (R^2^ = 0.9972−0.9999). The results were expressed in mg of compound per kg of dry GP. All analyses were performed in triplicate.

In order to evaluate the effect of each digestion process (gastric and intestinal) on the phenolic groups (anthocyanins, phenolic acids, flavonoids, and stilbenes), the recovery index (RI) was calculated according to the equation [33,34]: RI (%) = (A/B) × 100 where A and B, expressed as mg/kg dry weight, are the phenolic content quantified in each tested extract at each digestion process, and the phenolic content in the tested extract before digestion, respectively.

### 2.7. Microorganisms and Culture Conditions

*Lactiplantibacillus plantarum*, *Escherichia coli*, *Bacillus megaterium*, and *Listeria monocytogenes* belonged to the Culture Collection of Di.S.S.P.A of the University of Bari.

*L. plantarum* was propagated in MRS broth (Oxoid Ltd., Basingstoke, Hampshire, UK) for 24 h at 37 °C, *E. coli*, *B. megaterium*, and *L. monocytogenes* were propagated in Luria Bertani (LB) broth (Oxoid Ltd., Hampshire, UK) for 24 h at 30 °C.

### 2.8. Evaluation of Effects Exerted by Antioxidants, Prebiotics, and Pathogens on Probiotic Growth

Experimentation aimed at evaluating the above-mentioned strain growth on fecal media were made in combinations with digested GP extracts, showing major antioxidant activity. To evaluate the capabilities of the selected probiotics to grow in co-culture under conditions simulating the intestinal ecosystem, fecal extracts were used as model media [35]. Fecal media consisted of pooled feces from three healthy subjects freshly collected in a sterile stool container. Feces were homogenized in sterile bags with filters, using a stomacher (BagMixer, Interscience International, Roubaix, France) for 3 min. Feces were used at 25% (*w*/*v*) in NaCl-solution at 0.9%. The filtered suspension of feces was added with dipotassium phosphate (2 g/L), sodium acetate (5 g/L), triammonium citrate (2 g/L), magnesium sulfate (0.2 g/L), manganese sulfate (0.05 g/L), tween 80 (polysorbate, 1 mL/L), glucose (2 g/L), and bile salts (0.5 g/L), and were then sterilized at 121 °C for 15 min. After sterilization, cysteine HCl 0.5 g/L and hemin 0.02 g/L (previously sterilized by cooling filtration) were added to constitute the fecal medium. The obtained fecal media were singly inoculated with digested GP extract (at a concentration of 1 g/100 g) and with the selected probiotics and pathogens at a cell density of 7 log CFU mL^−1^ and incubated at 37 °C for 24 h. After 24 h, cell density was estimated by pour-plating in de Man, Rogosa, and Sharpe (MRS) (Oxoid, Basingstoke, Hampshire, England), and Luria-Bertani (LB) containing per liter: 10 g of tryptone, 5 g of yeast extract, 10 g of sodium chloride (all ingredients were purchased by Oxoid) agar adjusted at pH 7, incubated at 37 °C for 48 h. The pH was estimated by pH meter (Denver Instrument, Bohemia, NY, USA).

To assess the possible negative or positive effects of GP antioxidants on different intestinal microorganism growths, including probiotics, microbiological tests were conducted to select prebiotics and pathogens, even in the absence of GP antioxidants.

### 2.9. Statistical Analysis

Significant differences between the values of all parameters were determined at *p* < 0.05, according to the analysis of variance (ANOVA) followed by the Tukey’s honestly significant difference (HSD) test and Fisher least significant difference (LSD) test for multiple comparisons. The statistical analysis was performed by the Minitab Statistical Software (Minitab Inc., State College, PA, USA).

## 3. Results and Discussion

### 3.1. Antioxidant Activity of GP Extracts

Several studies conducted on different grape varieties showed how GP contains high antioxidant activity [34,36,37] associated with polyphenol content and composition [38]. To evaluate the effects of GP extracts on the gastrointestinal digestion process, the antioxidant activity was determined. The antioxidant activity of aqueous and hydroalcoholic GP extracts before and after the in vitro simulated gastric and intestinal digestion are shown in Figure 1. Of note, the antioxidant activity of extracts determined by the ABTS and DPPH assays showed a similar trend, confirming the correlation between these two assays [39].

Regarding GP aqueous extracts (Figure 1A,B), those sampled after intestinal digestion (-i) showed the highest antioxidant activity, followed by gastric-digested (-g) and undigested. The increase of antioxidant activity as a result of gastrointestinal digestion processes may derive from the hydrolysis and release of lower molecular weight metabolites [40]. Specifically, the change of the pH from 2.2 to 6.0 during gastrointestinal digestion determined an increase of antioxidant activity, as shown in previous studies [41,42]. These results are in line with Del Pino-Garcìa et al. (2016) [43], who found an increase of red wine pomace antioxidant activity after gastrointestinal digestion.

Among GP extracts subjected to intestinal digestion, the highest antioxidant activity was recorded in control (ACW) and oak (AQW) samples for the Aglianico cultivar, and in toasted (NTW) and oak (NQW) samples for the Nero di Troia cultivar. Although the gastric-digested GP extracts showed lower antioxidant activity than the intestinal-digested extracts, they were significantly higher compared to undigested samples. Specifically, regarding GP gastric-digested extracts, the highest antioxidant activity was detected in ACW and AQW sample for Aglianico cultivar, and in NBTW, NTW, and NQW samples for Nero di Troia cultivar. A similar trend was found for the undigested GP extracts: for Aglianico (antioxidant activity of ACW > AQW, ABTW, and ATW) and for Nero di Troia (antioxidant activity of NBTW, NTW, and NQW > NCW). Overall, for the undigested extracts, the addition of oak chips and vine shoots decreased the antioxidant activity in Aglianico samples, whereas it increased in the Nero di Troia samples.

Generally, the hydroalcoholic extracts (Figure 1C,D) showed higher antioxidant activity than the aqueous ones, with an average value of 120 µmol TE/g. Indeed, it is known that the extraction yield of phenolic compounds depends on the extraction technology chosen [44] and the extraction efficiency is affected by solvent type. Using hydroalcoholic solutions (polar protic media) as extraction solvents, polyphenols are easily solubilized due to their polar nature [45].

Of note, a significant decrease in antioxidant activity after the vitro gastrointestinal digestion for hydroalcoholic extracts was shown. This result was in line with others [34,46] that found a decrease in antioxidant activity in hydroalcoholic GP extracts after the in vitro simulated digestion. Among the undigested GP extracts, ACS showed a significantly higher antioxidant activity than AQS. By contrast, minor differences in antioxidant activity were found for GP extracts of the Nero di Troia cultivar. Additionally, concerning different GP extracts, samples subjected to gastric and intestinal digestions showed the same trend of those not subjected to digestion: ACS had higher antioxidant activity than ABTS, ATS, and AQS for Aglianico, whereas NQS showed a slight prevalence on NCS, NBTS, and NTS for Nero di Troia. Evident differences in antioxidant activity were found between aqueous-digested and hydroalcoholic extracts and may be due to the application of the protocol useful in evaporating and lyophilizing the hydroalcoholic extracts prior to the in vitro gastrointestinal digestion. As largely documented, a minor stability of polyphenols to temperature, light, and lyophilization conditions may result from exposure to high temperatures and freeze-drying steps that inevitably reduce their antioxidant activity [47,48,49]. In line with this, our results highlight a decrease of antioxidant activity in ACgS and AciS compared to ACS (Figure 1C,D).

### 3.2. Quantitative UHPLC-DAD Analysis of Phenolic Compounds

Based on the antioxidant activity results, only the aqueous and hydroalcoholic GP extracts with the highest antioxidant activities (AC, AQ, NC, and NQ) were screened and subsequently used for the UHPLC-DAD characterization. The content of quantified phenolic compounds by UHPLC-DAD in the selected GP extracts was divided into anthocyanins, phenolic acids, flavonoids, and stilbenes classes (Table 2), and the RI (%) of each phenolic group is shown in Table 3. The largest contribution for total polyphenols was found by anthocyanins, followed by phenolic acids, flavonoids, and stilbenes. Even though a great aliquot of anthocyanins is transferred from the skins of red grapes to the must during the maceration step in winemaking [10], GP still retains significant amounts of these compounds [7,28], with well-known beneficial properties [50,51]. GP undigested extracts of Aglianico cultivar showed a higher concentration of total phenolic compounds when compared with the ones from the Nero di Troia cultivar. These results indicate a peculiar role of the cultivar that exerts a strong impact on the phenolic compound content in GP at the end of the winemaking process, due to the genome, soil and climatic conditions, grape maturation, and (mainly) the winemaking technology [7].

In line with the results of antioxidant activity, the GP undigested hydroalcoholic extracts (ACS, AQS, NCS, and NQS) showed higher concentrations of total polyphenols than the aqueous ones (ACW, AQW, NCW, and NQW). In addition, ACW and AQW lacked stilbenes, NCW and NQW also lacked flavonoids as well as stilbenes. Moreover, control samples (-C) showed higher values of total polyphenols than those added with oak chips (-Q). Concerning the concentration of total phenols, the simulated in vitro digestion processes had different effects, depending on the type of extract (aqueous or hydroalcoholic). Specifically, the gastric digestion process of aqueous extracts—except for NQgW—caused an increase of total phenol concentration (i.e., from 1176.9 mg/kg in ACW to 3811.7 mg/kg in ACgW), whereas intestinal digestion caused a decrease of total phenol concentration (i.e., from 3811.7 mg/kg in ACgW to 630.0 mg/kg in ACiW). As shown in Table 3, the RI of the total phenol content decreased in all aqueous extracts after intestinal digestion (RI 53.5, 14.9, 15.9, and 14.8%, respectively). Of note, RI increased after intestinal digestion for phenolic acid and flavonoids in ACiW (311.3 and 1260.7%, respectively).

The highest contribution of total phenol concentration increase came from anthocyanins. In fact, the literatures results show major “stability” of anthocyanins in gastric pH conditions rather than intestinal pH conditions [52,53].

Conversely, the total phenol content of hydroalcoholic (-S) extracts decreased from undigested to gastric (-g) and intestinal (-i)-digested samples. In fact, total contents of phenolic compounds decreased after the gastric and intestinal digestions for all samples (RI of 23.7, 17.8, 7.1, and 21.5% for gastric extracts, RI of 1.0, 0.6, 0.6, and 0.1% for intestinal extracts, respectively) (Table 3).

For example, in the ACS sample, the gastric digestion process caused a reduction of about 76% in total polyphenols; this is in line with a previously published study showing how the in vitro gastrointestinal process reduced the polyphenol and anthocyanin contents in Merlot GP extracts by 49% and 15%, respectively [54]. GP-digested extracts reached higher values of total polyphenols for Aglianico samples compared with the Nero di Troia ones, as observed for the undigested samples (Table 2).

Extracts subjected to intestinal (-i) digestion showed lower total polyphenol content than gastric (-g) digestion extracts. The differences between the aqueous and hydroalcoholic extracts flattened out, with a slight predominance of the latter. In particular, anthocyanin levels decreased after intestinal digestion due to the pH variations between the stomach (pH 1.5–3.5) and intestine (pH 6.7–7.4). This could be explained by the total biotransformation of anthocyanins into low molecular weight molecules, such as phenolic acids and catechol, which improve their bioaccessibility and bioavailability [51]. Moreover, anthocyanins could form complexes with large molecular weight constituents preventing the crossing of the dialysis membrane [55]. This evidence disagrees with the increase of antioxidant activity of the same aqueous extracts and could be explained by the production of metabolites with lower molecular weight during the in vitro gastrointestinal digestion not detected in the UHPL-DAD analysis [56].

### 3.3. Growth of Probiotics and Pathogens in the Presence of GP Antioxidants

The intestinal aqueous-digested GP extract with the highest antioxidant activity (AQiW) was used to evaluate the in vitro effects of the combination of antioxidants, probiotics, and pathogens. As a control, the same theses were also tested in the absence of AQiW. Therefore, to evaluate the effect of polyphenolic compounds on the growth of probiotic (*L. plantarum*) and pathogenic (*E. coli*, *B. megaterium,* and *L. monocytogenes*) microorganisms, the AQiW at a concentration of 1 g/100 g was used. Both stimulatory and inhibitory effects were observed in comparison to the positive control.

Compared to the controls, a low decrease in pH was indiscriminately observed for *L. plantarum*, *E. coli*, *B. megaterium*, and *L. monocytogenes* (data not shown) after 24 h of incubation in the presence—and absence—of GP.

Moreover, as confirmed by cell density evaluation, after 24 h of incubation, *L. plantarum* showed an increase in the presence of GP (Figure 2). In addition, when a density of 7 log CFU mL^−1^ was modified to 4.40, 4.30, and 4.50 log CFU mL^−1^, a decrease of *E. coli*, *B. megaterium*, and *L. monocytogenes* was observed when GP was inoculated with *L. plantarum*. *E. coli* grew (8.45 log CFU mL^−1^) only when it was inoculated with *L. plantarum*, whereas *B. megaterium* and *L. monocytogenes* (at 4.46 and 4.60 log CFU mL^−1^, respectively) decreased. These results confirmed the antimicrobial activity against pathogens due to the combination of GP and *L. plantarum*.

To date, several studies have been published pertaining to the influence of *Lactiplantibacillus* genera-derived polyphenols on LAB growth and viability. Bacteria are capable of metabolizing these compounds as substrates and grow in their presence [57]. In GP, antioxidants positively affect bacteria metabolism, and enhance nutrient consumption, as in the case of sugars [58,59,60]. Generally, as observed in the fermentation of green table olives, a high concentration of polyphenols could inhibit or slow down the growth of LAB, but a modification of pH could also improve the lactic fermentation [61]. Similarly, the modification of pH during gastrointestinal digestion contributes to the metabolization of phenolics acids and their utilization for *L. plantarum*. Analogous findings were observed for another species of *Lactiplantibacillus*, i.e., *L. acidophilus* was not inhibited by phenolic compounds from GP [57]. It is important to consider that the type of polyphenol, its form, concentration, and the susceptibility of bacteria strain, can modulate the effect of polyphenolic compounds [62]. Researchers have largely investigated the effects of polyphenols on gut microbiota, studying the inhibition and stimulation of specific microorganisms in relation to phenolic compounds, such as anthocyanins [63]. As reported by Lee et al. (2006) [64], polyphenols from tea were able to inhibit pathogenic bacteria (*Clostridium perfringens*, *Clostridium difficile* and *Bacteroides*) and, at the same time, the growth of probiotic bacteria, such as *Bifidobacterium* and *Lactiplantibacillus* sp. [65,66]. In accordance with our results, Salih et al. (2000) [67] reported that *L. plantarum* is able to grow in the presence of phenolic acids. Since the time of their discovery, due to their antioxidant properties and inhibitory effects on various microorganisms, phenolic compounds have attracted much attention.

As mentioned by Peixoto et al. (2018) [19], a strong correlation was observed between GP polyphenol levels and the growth of Gram-negative bacteria, such as *E. coli*. Several studies have showed the antimicrobial activity of wine and grapes against pathogens, such as *Staphylococcus aureus*, *E. coli*, *Candida albicans*, and *Listeria monocytogenes* [68,69,70]. However, the phenolic compounds from different parts of grapes have been found to exert different antimicrobial effects. In particular, fermented pomace showed significantly higher antimicrobial activity than whole fruit grape extracts [71].

## 4. Conclusions

The hydroalcoholic extracts of GP have greater antioxidant activity than aqueous extracts, this is due to the higher concentrations of anthocyanins, phenolic acids, flavonoids, and stilbenes. Regarding the simulated in vitro gastric and intestinal digestion, the antioxidant activity of aqueous extracts increases after intestinal digestion, whereas the one relative to hydroalcoholic extracts dramatically decreases. Despite a lower global antioxidant activity marking aqueous extracts, these inhibited the growth of the tested pathogens and enhanced the growth of probiotic bacteria. In general, polyphenol extracts were effective against Gram-positive (*B. megaterium* and *L. monocytogenes*) and Gram-negative (*E. coli*) bacteria. Notably, our results confirm that extracts from red winemaking GP are an important source of antioxidant compounds that are bioaccessible after simulating gastrointestinal digestion processes. These extracts could be used as functional food ingredients, or in the formulation of food supplements and cosmetics. This finding shows the potential application of phenolic compounds in maintaining shelf life and improving the safety of ready-to-eat food products.

## Figures and Tables

**Figure 1 antioxidants-11-00567-f001:**
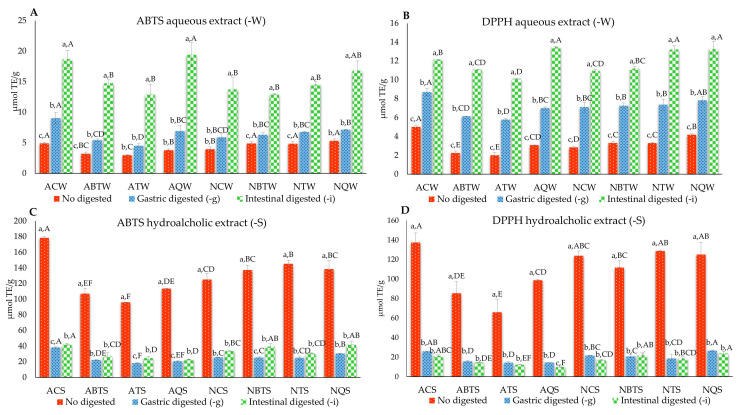
Antioxidant activity of aqueous (**A**,**B**) and hydroalcoholic (**C**,**D**) GP extracts before and after simulating the gastric (-g) and intestinal (-i) digestion. Data are expressed as mean values ± standard deviation (SD). Different lower-case letters indicate significant differences among the same extracts versus different samples (*p* < 0.05, one-way ANOVA and Tukey’s HSD test). Capital letters indicate a significant difference (*p* < 0.05, one-way ANOVA and Tukey’s HSD test) when comparing different GP samples against the same conditions (undigested, gastric-digested, or intestinal-digested). For sample codes, see Table 1.

**Figure 2 antioxidants-11-00567-f002:**
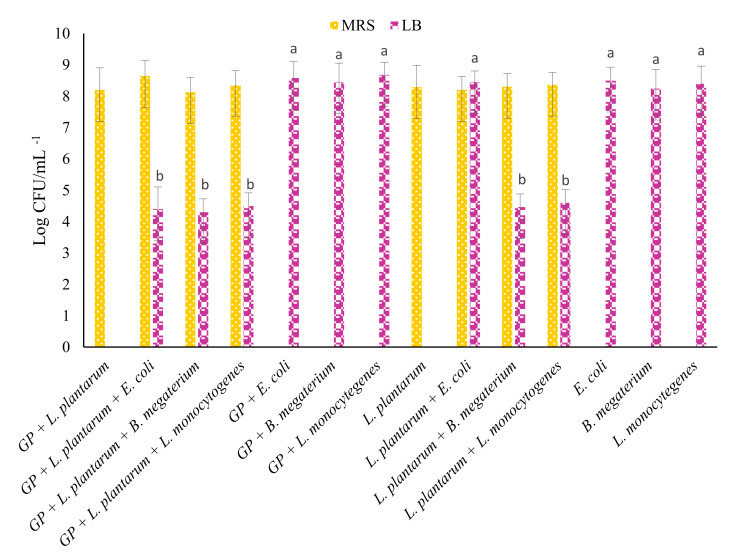
Viable cell count (log CFU mL^−1^) of probiotics and pathogens after 24 h of growth at 37 °C in MRS and LB. Data are presented as the average of biological triplicates ± SD. Different letters indicate statistically significant differences among LB samples. Different letters (^a–b^) indicate significant differences (*p* < 0.05) in LB count according to Tukey’s HSD test. Abbreviations: GP, grape pomace extract; LB, Luria Bertani; MRS, De Man, Rogosa and Sharpe.

**Table 1 antioxidants-11-00567-t001:** Samples analyzed and acronyms used throughout the text.

Variety	Thesis	Acronyms
*Aglianico*	Control	AC
	Control—gastric-digested	ACg
	Control—intestinal-digested	ACi
	Vine shoots boiled and toasted	ABT
	Vine shoots boiled and toasted—gastric-digested	ABTg
	Vine shoots boiled and toasted—intestinal-digested	ABTi
	Vine shoots toasted	AT
	Vine shoots toasted—gastric-digested	ATg
	Vine shoots toasted—intestinal-digested	ATi
	Oak chips	AQ
	Oak chips—gastric-digested	AQg
	Oak chips—intestinal-digested	AQi
*Nero di Troia*	Control	NC
	Control—gastric-digested	NCg
	Control—intestinal-digested	NCi
	Vine shoots boiled and toasted	NBT
	Vine shoots boiled and toasted—gastric-digested	NBTg
	Vine shoots boiled and toasted—intestinal-digested	NBTi
	Vine shoots toasted	NT
	Vine shoots toasted—gastric-digested	NTg
	Vine shoots toasted—intestinal-digested	NTi
	Oak chips	NQ
	Oak chips—gastric-digested	NQg
	Oak chips—intestinal-digested	NQi

The last letter for each sample acronym indicates the different type of extraction (W or S). W stands for aqueous extract; S stands for hydroalcoholic extract.

**Table 2 antioxidants-11-00567-t002:** Quantified sample content (mg/kg dry weight ± SD) of the main classes of phenolic compounds by the UHPLC-DAD analysis.

	Samples	Anthocyanins (mg/kg Dry Weight)	Phenolic Acid (mg/kg Dry Weight)	Flavonoids (mg/kg Dry Weight)	Stilbenes (mg/kg Dry Weight)	TOTAL (mg/kg Dry Weight)
Undigested	ACW	1109.9 ± 2.8 ^j^	65.7 ± 10.1 ^hij^	1.3 ± 0.2 ^g^	/	1176.9 ± 13.1 ^j^
	AQW	753.1 ± 37.8 ^jk^	43.7 ± 0.2 ^mn^	2.8 ± 0.2 ^g^	/	799.6 ± 38.2 ^ij^
	NCW	444.0 ± 22.1 ^klm^	17.7 ± 1.3 ^r^	/	/	461.6 ± 23.4 ^klmn^
	NQW	336.7 ± 0.1 ^klm^	60.0 ± 9.8 ^ijk^	/	/	396.7 ± 9.7 ^klmn^
	ACS	68,079.2 ± 685.0 ^a^	661.7 ± 21.1 ^c^	230.0 ± 23.4 ^c^	69.5 ± 12.8 ^a^	69,040.4 ± 669.9 ^a^
	AQS	30,512.8 ± 108.5 ^b^	604.7 ± 13.7 ^d^	235.7 ± 0.5 ^c^	42.4 ± 0.2 ^b^	31,395.6 ± 95.6 ^b^
	NCS	23,108.3 ± 240.1 ^c^	1160.5 ± 13.8 ^a^	347.3 ± 28.9 ^b^	33.9 ± 0.2 ^bc^	24,650.0 ± 255.4 ^c^
	NQS	21,631.6 ± 1140.9 ^d^	1119.5 ± 1.8 ^b^	486.3 ± 12 ^a^	26.4 ± 0.9 ^c^	23,263.7 ± 1151.1 ^d^
Gastric-digested (-g)	ACgW	3783.6 ± 54.1 ^h^	19.8 ± 3.5 ^qr^	8.4 ± 1.1 ^fg^	/	3811.7 ± 56.5 ^h^
	AQgW	1826.0 ± 138.5 ^i^	110.6 ± 6.9 ^f^	8.8 ± 0.8 ^fg^	/	1945.4 ± 146.2 ^i^
	NCgW	743.8 ± 4.3 ^jk^	69.3 ± 0.7 ^hi^	/	/	813.1 ± 3.6 ^jk^
	NQgW	269.6 ± 22.5 ^klm^	36.0 ± 0.8 ^nop^	/	/	305.6 ± 23.3 ^lmn^
	ACgS	16,308.3 ± 24.4 ^e^	30.4 ± 6.4 ^opq^	55.5 ± 17.9 ^d^	/	16,394.2 ± 48.7 ^e^
	AQgS	5502.1 ± 109.1 ^f^	42.1 ± 1.9 ^mno^	31.1 ± 1.6 ^e^	/	5575.3 ± 105.7 ^f^
	NCgS	1725.8 ± 0.04 ^i^	16.2 ± 2.9 ^r^	5.1 ± 1.1 ^g^	/	1747.2 ± 4.1 ^i^
	NQgS	4918.8 ± 396.5 ^g^	56.1 ± 3.2 ^jkl^	22.8 ± 1.5 ^ef^	/	4997.8 ± 401.2 ^g^
Intestinal-digested (-i)	ACiW	409.2 ± 36.3 ^klm^	204.5 ± 1.4 ^e^	16.4 ± 1.0 ^efg^	/	630.0 ± 38.7 ^klm^
	AQiW	68.5 ± 1.1 ^m^	46.8 ± 0.3 ^lmn^	4.1 ± 0.02 ^g^	/	119.4 ± 0.9 ^n^
	NCiW	/	73.5 ± 6.3 ^h^	/	/	73.5 ± 6.3 ^n^
	NQiW	/	58.7 ± 9.3 ^ijkl^	/	/	58.7 ± 9.3 ^n^
	ACiS	586.4 ± 18.0 ^kl^	86.0 ± 4.6 ^g^	14.9 ± 1.0 ^efg^	/	687.2 ± 23.6 ^kl^
	AQiS	148.5 ± 1.8 ^lm^	30.9 ± 5.4 ^opq^	6.5 ± 0.2 ^fg^	/	185.9 ± 3.4 ^mn^
	NCiS	101.5 ± 1.7 ^lm^	52.1 ± 2.0 ^hijklm^	/	/	153.7 ± 3.7 ^mn^
	NQiS	/	28.4 ± 0.4 ^pqr^	/	/	28.4 ± 0.4 ^n^

/ = analyzed but not detected; all values are means ± SD belonging to the three replicate measurements. Statistically significant means (*p* ≤ 0.05, one-way ANOVA and Fisher LSD test) within each column are indicated by letters (^a–r^). For sample codes, see Table 1.

**Table 3 antioxidants-11-00567-t003:** Recovery Index % (RI) of each phenolic group (anthocyanins, phenolic acid, flavonoids, stilbenes and total) in samples extracts before (undigested) and after in vitro digestion (gastric- and intestinal-digested).

	Samples	RI Anthocyanins (%)	RI Phenolic Acid (%)	RI Flavonoids (%)	RI Stilbenes (%)	RI Total (%)
Undigested	ACW					
Gastric	ACgW	340.9	30.1	643.8	0	323.9
Intestinal	ACiW	36.9	311.3	1260.7	0	53.5
Undigested	AQW					
Gastric	AQgW	242.5	253.1	318.5	0	243.3
Intestinal	AQiW	9.1	107.1	147.6	0	14.9
Undigested	NCW					
Gastric	NCgW	167.5	392.1	0	0	176.1
Intestinal	NCiW	0	415.9	0	0	15.9
Undigested	NQW					
Gastric	NQgW	80.1	59.9	0	0	77.0
Intestinal	NQiW	0	97.9	0	0	14.8
Undigested	ACS					
Gastric	ACgS	24.0	4.6	24.1	0	23.7
Intestinal	ACiS	0.9	13.0	6.5	0	1.0
Undigested	AQS					
Gastric	AQgS	18.0	7.0	13.2	0	17.8
Intestinal	AQiS	0.5	5.1	2.7	0	0.6
Undigested	NCS					
Gastric	NCgS	7.5	1.4	1.5	0	7.1
Intestinal	NCiS	0.4	4.5	0	0	0.6
Undigested	NQS					
Gastric	NQgS	22.7	5.0	4.7	0	21.5
Intestinal	NQiS	0	2.5	0	0	0.1

For sample codes, see Table 1.

## Data Availability

Data are contained within the article.

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
