# Peer review of "Effects of Grape Pomace Polyphenols and In Vitro Gastrointestinal Digestion on Antimicrobial Activity: Recovery of Bioactive Compounds"

_antioxidants, 2022, doi:10.3390/antiox11030567_

Round 1
Reviewer 1 Report
The study is interesting described but some observations have to be done:
245-252 The results of the study were compared with other studies performed in different conditions and using other extraction methods for interest compounds. You mentioned as a possible explanation for increasing antioxidant activity, the hydrolysis and release of the phenolic compounds. But why this explanation is not available for alcoholic extracts, because your references used in the text describe alcoholic extracts.
You have to find references in which it was studied the in vitro digestion process using alcoholic or aqueous extracts.
Table 2. What represents the 3 parts of the table? It is not mentioned. Why didn't you use a recovery index for phenolics?
Fig 1. What represents the error bars? It is not mentioned. Please verify the data from the figure. Are you sure there is no statistical differences between AQW no dig vs gastric dig (Fig 1 A)?
For capital letters, if you compare results 3 by 3, for ex ABTW (fig 1 A) what is the meaning of "D"? and in ACW, gastric dig and int dig have "A". Are you sure?
Reviewer 2 Report
This work presents a fairly complete study of the phenolic content and antioxidant activity of aqueous and hydroalcoholic extracts from grape pomace before and after being subjected to simulated in vitro gastrointestinal digestion. It has been completed with the study of the antimicrobial activity against different pathogenic microorganisms of the extracts that present the highest antioxidant activity. The work is well written, the bibliography is appropriate and relevant and the conclusions are well supported by the results obtained. However, in my opinion some results need further discussion.
The authors attribute the increase of antioxidant activity of aqueous extracts after gastrointestinal digestion to the hydrolysis and release of phenolic compounds from the matrix (lines 247-248). Certainly there are previous references in the literature on this fact, although it generally occurs in gastric digestion (acidic pH). The authors then verify that the total phenolic content in these extracts decreases after gastrointestinal digestion, which is in disagreement with the increase in antioxidant activity, thus attributing this increase to the production of metabolites with a non-phenolic structure (lines 326- 327). This is a contradiction.
On the other hand, the undigested hydroalcoholic extracts showed an antioxidant activity significantly higher than that of the digested extracts, both gastric and intestinal ones. Similar results have also been previously reported by other authors. However, the authors do not analyze why the trend in aqueous and hydroalcoholic extracts is totally opposite, despite the fact that the matrix is ​​the same.
Other comments
. Abstract (line 14). Bio-product or by-product?
. Introduction (line 67) and Section 3.3. (line 337). Please change ‘prebiotic’ to ‘probiotic’.
. Section 2.6. What standards were used for the semi-quantification of flavonoids, phenolic acids, anthocyanins and stilbenes, respectively? The authors could easily have determined the total phenolic content (TPC) and the total flavonoid content (TFC) in order to complete the results obtained by the UHPLC-DAD analysis.
Round 2
Reviewer 1 Report
All recommendations were respected